# Surveillance of Antimicrobial Resistance and Multidrug Resistance Prevalence of Clinical Isolates in a Regional Hospital in Northern Greece

**DOI:** 10.3390/antibiotics12111595

**Published:** 2023-11-05

**Authors:** Maria Tsalidou, Theodouli Stergiopoulou, Ioannis Bostanitis, Christina Nikaki, Kalypso Skoumpa, Theofani Koutsoukou, Paraskevi Papaioannidou

**Affiliations:** 1Laboratory of Microbiology, General Hospital of Katerini, 60100 Katerini, Greece; 21st Department of Pharmacology, School of Medicine, Faculty of Health Sciences, Aristotle University of Thessaloniki, 54124 Thessaloniki, Greeceppap@auth.gr (P.P.)

**Keywords:** antimicrobial resistance, antimicrobial surveillance, multidrug resistance, MDR, XDR, PDR, VRE, MRSA, hospital infections, Greece

## Abstract

Antimicrobial resistance is a global health threat resulting in increased morbidity and mortality. This retrospective study aimed to estimate antimicrobial susceptibility and multidrug resistance prevalence of clinical isolates in a regional hospital in Northern Greece during the last 6 years by analyzing the annual reports of the Laboratory of Microbiology. A total of 12,274 strains of certain bacteria were isolated from both hospitalized and ambulatory patients from biological products, mainly urine (range 63–78% during the study period). *E. coli* was the most frequent pathogen found (37.4%). A significant increase in the number of the main pathogens causing hospital-acquired infections (*Klebsiella pneumoniae*, *Acinetobacter baumannii*, *Pseudomonas aeruginosa*, *Enterococcus faecium*) was found in the time period of 2021–2023 compared to 2018–2020 (*p* < 0.0001). In total, 1767 multidrug-resistant bacterial strains were isolated, most of them belonging to *Acinetobacter baumannii* (36.4%) and *Klebsiella pneumoniae* (39.6%), and were located in the intensive care unit (ICU) (59.8%). Extensively drug resistance (XDR) and pan drug resistance (PDR) were significantly higher in 2021–2023 than in 2018–2020 (XDR: 641/1087 in 2021–2023 vs. 374/680 in 2018–2020 and PDR: 134/1087 in 2021–2023 vs. 25/680 in 2018–2020, *p* < 0.0001), resulting in an urgent need to establish certain strategies in order to eliminate this threatening condition.

## 1. Introduction

Antimicrobial resistance is characterized as a “silent pandemic” by the global scientific community and is considered as one of the biggest threats to public health nowadays [1]. The World Health Organization (WHO) talks about a “post-antibiotic era” in which there will be no antibiotics suitable to deal with pan-resistant microbes [2]. In fact, it is estimated that by 2050 the problem of antimicrobial resistance will cause more deaths than cancer and diabetes combined [3].

Europe seems to share the same problem, as in 2015 the European Centre for Disease Prevention and Control (ECDC) recorded 33,110 deaths due to antibiotic-resistant microbes while predicting that by 2050 an economic crisis similar to that of 2008 will appear, due to the increased cost of hospitalizations and the decrease in productivity, as a result of antimicrobial resistance [4].

Regarding Greece, the prevalence of hospital infections is estimated to be about 10%, which means that 66,487 hospitalized patients suffer from some nosocomial pathogens annually [5]. The most frequently encountered pathogens are the following: *Acinetobacter baumannii*, *Escherichia* Coli, *Klebsiella pneumoniae*, *Pseudomonas aeruginosa*, *Staphylococcus aureus*, and *Enterococcus faecium* and *faecalis* [6]. A major problem is the resistance of *Klebsiella pneumoniae* to third-generation cephalosporins and carbapenems (66.5% vs. 31.3% in the EU and 58.3% vs. 7.9% in the EU, respectively), the resistance of *Pseudomonas aeruginosa* to carbapenems and fluoroquinolones (48.9% vs. 16.5% in the EU and 46.8% vs. 18.9% in the EU, respectively), and the increasing multiresistance of *Acinetobacter baumannii* to cefepime, carbapenems, and aminoglycosides in both ICUs and clinical wards [5,6].

In Greece, there are only a few studies on the surveillance of antimicrobial resistance [5,6,7]. Our study aimed to estimate the antimicrobial susceptibility and multidrug resistance prevalence of certain clinical isolates in a regional hospital in Northern Greece during the last 6 years and to compare the difference between the 3-year time periods of 2018–2020 and 2021–2023. We expect that our study will lead to more effective antimicrobial treatment and better clinical outcomes.

## 2. Results

### 2.1. Six-Year Surveillance of Clinical Isolates’ Susceptibility to Certain Antibiotics

#### 2.1.1. Types of Clinical Isolates

During the 6-year period of 2018–2023 (January 2018 through June 2023), 12,274 bacterial strains were isolated from both hospitalized and community patients in the General Hospital of Katerini (Table 1).

Most clinical strains were isolated from biological products during the peak of the COVID-19 pandemic years (n = 2321 and n = 2658 in 2020 and 2021, respectively). Gram-positive cocci (*Enterococcus faecalis, Enterococcus faecium, Staphylococcus aureus*) were notably fewer than Gram-negative bacilli (*Escherichia coli, Klebsiella pneumoniae, Enterobacter species, Acinetobacter baumannii, Pseudomonas aeruginosa*) (10.5 % vs. 74.8%) while other pathogens were estimated to be about 14.7%. *E. coli* was the most frequent pathogen found (37.4 %), followed by *Klebsiella pneumoniae* (18.2%).

In the COVID-19 pandemic years, a significant increase in the number of main pathogens causing hospital-acquired infections (*Klebsiella pneumoniae*, *Acinetobacter baumannii*, *Pseudomonas aeruginosa*) was found (Table 1).

Most isolated bacterial strains were obtained from urine cultures (ranging from 63 to 78%), followed by blood cultures (6–10%) and rectal and pharyngeal specimens (2–7%) (Table 2). The number of urine strains (deriving mainly from community specimens) was decreased during the peak of the COVID-19 pandemic in 2021, while blood, pharyngeal, rectal, and catheter specimens were increased, reflecting a long hospitalization due to COVID-19 infection.

#### 2.1.2. Antimicrobial Resistance of Clinical Isolates

Detailed results on antimicrobial resistance are depicted in Figure 1, Figure 2, Figure 3, Figure 4, Figure 5, Figure 6 and Figure 7.

##### Gram-Positive Cocci

The resistance of *Enterococcus faecalis* to ampicillin (AMP) decreased from 2018 to 2023 (4.5% to 0%, Figure 1A, *p* < 0.05). On the contrary, the incidence of isolated *Enterococcus faecium* strains resistant to ampicillin was very high during all years, ranging from 92.3 to 100%. Resistance to linezolid (LNZ) tended to be higher in *Enterococcus faecalis* than in *Enterococcus faecium* both in 2018–2020 and 2021–2023 (1.9–6.0% vs. 0.0–3.8%, respectively, *p* > 0.05). Furthermore, an increasing trend of vancomycin-resistant (VRE) *Enterococcus faecium* strains was observed in 2021–2023 (34.6–57.5% in 2021–2023 vs. 11.5–59% in 2018–2020, respectively, *p* > 0.05). Resistance to teicoplanin (TEC) tended to be higher in the second time period (25–57.7% in 2021–2023 vs. 3.8–27.6% in 2018–2020, *p* > 0.05). These results are depicted in Figure 1B and Figure 2.

As for *Staphylococcus aureus*, all strains were excessively sensitive (100%) to linezolid (LNZ), daptomycin (DAP), and teicoplanin (TEC) in both time periods. Resistance to (VAN) vancomycin was also low (0% in 2021–2023 vs. 2.7–6.8% in 2018–2020) while resistance to ciprofloxacin (CIP) was quite high, ranging from 13.5 up to 55.9% during both the 2018–2020 and 2021–2023 time periods, *p* > 0.05. The presence of methicillin-resistant *Staphylococcus aureus* (MRSA) strains, reported by cefoxitin (FOX) resistance, ranged—however, not significantly—during both time periods, being 16.2% in 2019 and 66.1% in 2021, *p* > 0.05 (Figure 1C).

##### Enterobacterales

The antimicrobial-resistant patterns of *E. coli* were similar in both examined time periods (2018–2020 and 2021–2023, Figure 3A).

All strains had high sensitivity to most antibiotic categories, mainly carbapenems, colistin, amikacin, third-generation cephalosporins, and piperacillin/tazobactam (PTZ). The highest resistance rates were observed in amoxicillin/clavulanic acid (AMC) (mean 58–60%) as well as to fluoroquinolones (CIP, ciprofloxacin; LVX, levofloxacin), being 29.9% (2021) and 66.7% (2018) for LVX and approximately 28–30% for CIP (Figure 3A).

*Klebsiella pneumoniae* sensitivity to many antibiotic classes tended to be reduced from 2018 to 2023. Resistance to all the examined antibiotics tended to be high in both time periods, *p* > 0.05, except for amikacin (AMK) and colistin (COL). However, a rising trend was observed regarding the resistance to amikacin in the period of 2021–2023 (amikacin: 10.3–40.7% in 2018–2020 vs. 22.9–59.1% in 2021–2023, *p* > 0.05). As for colistin, a statistically significant rise in resistance during 2018–2023 was observed (15–19% in 2018–2020 vs. 19.9–36.1% in 2021–2023y, *p* < 0.05) (Figure 3B and Figure 4).

Antimicrobial resistance patterns showed similarities concerning the resistance of *Enterobacter* species (spp.) to certain drugs during 2018–2020 and 2021–2023. Although a limited number of *Enterobacter* spp. strains was isolated (both *E. cloacae* and *E. aerogenes* represented less than 1% of the isolates in both time periods), an obvious trend of increasing resistance of *Enterobacter cloacae* to all antibiotic classes except aminoglycosides (AMK) was observed (Figure 3C).

##### *Acinetobacter baumannii* and *Pseudomonas aeruginosa*

*Acinetobacter baumannii* showed extremely low susceptibility to all antibiotic categories, except colistin (COL). A further overall increasing resistance trend to all antibiotics was also observed in 2021–2023, including COL, as well (1.3–10.2% in 2021–2023 vs. 0.9–6% in 2018–2020, respectively); this difference was not significant (Figure 5A and Figure 6).

Similar antimicrobial resistance patterns were also observed in *Pseudomonas aeruginosa* in the two examined time periods, especially to third- and fourth-generation cephalosporins (CAZ, ceftazidime; FEP, cefepime), piperacillin/tazobactam (PTZ), and amikacin (AMK), while sensitivity was high for colistin (COL). Annual variations in carbapenems and fluoroquinolones resistance were registered, varying from 35–40% for imipenem/meropenem (IPM/MER) in 2018–2020 up to 40–45% in 2021–2023.

As for both fluoroquinolones, in particular, a small decreasing trend in antimicrobial resistance was reported (ciprofloxacin (CIP): 50% and levofloxacin (LVX): 70% in 2018–2020 vs. ciprofloxacin (CIP): 45% and levofloxacin (LVX): 55–60% in 2021–2023, *p* > 0.05) (Figure 5B and Figure 7).

### 2.2. Epidemiology and Prevalence of Multidrug-Resistant (MDR) Clinical Isolates in 2018–2020 and 2021–2023

In total, 1767 patients were found to be infected by multidrug-resistant (MDR) bacterial strains during 2018–2023. Of that, 1094 (62%) were males aged 72 (3–101) years and 673 (38%) were females aged 77 (1–98) years. A significant increase (93%, almost 2 fold) in MDR bacteria was observed in males compared to females (19%) in the time period of 2021–2023 compared to the time period of 2018–2020 (*p* < 0.0001, r = −0.115, Figure 8A). Another difference was that cases with MDR isolates in men were observed in a younger age during the second time period, 2021–2023, compared to 2018–2020 (two-way ANOVA, *p* < 0.0001, r = 0.062, Figure 8B).

MDR *Klebsiella pneumoniae* (700/1767, 39.6%) and *Acinetobacter baumannii* (643/1767, 36.4%) were the dominant MDR bacteria, followed by *Pseudomonas aeruginosa* (192/1767, 10.9%) and *Enterococcus faecium* (171/1767, 9.7%). The prevalence of bacteria with multidrug resistance is shown in Figure 9.

In particular, when bacterial strains were analyzed according to the kind of antimicrobial resistance (multidrug resistance, MDR; extensively drug resistance, XDR; pan drug resistance, PDR) and the presence of VRE and MRSA strains, XDR clinical isolates were the most prevalent ones (1015/1767, 57.4%), followed by MDR strains (570/1767, 32.3%) and PDR strains (159/1767, 9%). The prevalence of VRE and MRSA isolated bacterial strains was low (0.8% and 0.5%, respectively) (Figure 10).

The intensive care unit (ICU) harbored the majority of MDR isolates (1053/1760, 59.8%), followed by the department of internal medicine (377/1760, 21.4%). The prevalence of MDR in the community patients (hospital outpatient and emergency care units) was also high (211/1760, 11%) (Figure 11).

Most MDR isolates were obtained from urine cultures (673/1748, 38.5%) and from pharyngeal (355/1748, 20.3%) and rectal specimens (340/1748, 19.5%) and were collected in the ICU from blood- and catheter-related cultures (254/1748, 14.6%) and rarely from other biological products such as bronchial and injury specimens, etc. (Figure 12).

The comparison of MDR prevalence between 2018–2020 and 2021–2023 revealed a significant rise in *Klebsiella pneumoniae* (300/1767 in 2018–2020 vs. 400/1767 in 2021–2023, *p* = 0.002, r = 0.073), a non-significant rise in *Acinetobacter baumannii* (246/1767 in 2018–2020 vs. 397/1767 in 2021–2023, *p* > 0.05) and *Pseudomonas aeruginosa* (70/1767 in 2018–2020 vs. 122/1767 in 2021–2023, *p* > 0.05), respectively, and a significant rise in *Enterococcus faecium* (39/1767 in 2018–2020 vs. 132/1767 in 2021–2023) MDR isolates in the course of time (*p* < 0.0001, r = 0.105), as shown in Figure 13.

When analyzed for the type of antimicrobial resistance, MDR and PDR strains increased significantly during 2021–2023 when compared to 2018–2020 (MDR, *p* < 0.0001, r = 0.136; PDR *p* < 0.0001, r = 0.147). A non-significant rise in XDR strains in 2021–2023 when compared to these in 2018–2020 was also observed (*p* > 0.05). MDR strains were 274/680 in 2018–2020 and 296/1087 in 2021–2023, XDR strains were 374/680 in 2018–2020 and 641/1087 in 2021–2023, and PDR strains were 25/680 in 2018–2020 and increased to 134/1087 in 2021–2023. VRE and MRSA strains had a similar prevalence between the two time periods (VRE: 6/680 in 2018–2020 vs. 9/1087 in 2021–2023; MRSA: 1/680 in 2018–2020 vs. 7/1087 in 2021–2023) (Figure 14).

A non-significant increase in MDR strains located in the ICU was observed (661/1083) in 2021–2023 when compared to the number of isolates that presented in the ICU from 2018 to 2020 (392/677), *p* > 0.05. Smaller but significant increases in 2021–2023 were observed in the department of internal medicine (164/677 in 2018–2020 vs. 213/1083 in 2021–2023, *p* = 0.023, r = 0.054) and in the other wards (12/677 in 2018–2020 vs. 58/1083 in 2021–2023, *p* < 0.0001, r = 0.089). A non-significant rise in MDR strains in outpatient and emergency care units (91/677 in 2018–2020 vs. 120/1083 in 2021–2023, *p* > 0/05) was also observed (Figure 15).

## 3. Discussion

The WHO and the US Centers for Disease Control and Prevention (CDC) have characterized antimicrobial-resistant bacteria as a universally accelerating serious health threat [8,9]. Although there is not yet an integral, systematic, international surveillance of antimicrobial resistance [8], various reports indicate that more than 2 million infections due to antibiotic-resistant pathogens with 29,000 deaths occur in the United States per year, with a health care cost of more than USD 4.7 billion [9]. In Europe, data from national surveillance networks report over 33,000 deaths and 874,000 disability-adjusted life years due to hospital- and community-acquired infections every year, accounting for USD 1.5 billion in annual health costs [4,10]. The problem seems to occur in developed rather than developing countries, in which communicable infectious diseases remain the main cause of mortality [11], while recent data report increasing rates of MDR resistance as well [12].

Although antimicrobial resistance (AMR) genes exist naturally in the environment, the systematic empirical administration of broad-spectrum antibiotics in hospital settings has increased their presence in this area [13]. Therefore, in 2017, the WHO published a list of pathogens requiring the urgent discovery and administration of new antibiotic agents.

Within this list, certain pathogens named the ESKAPE (*Enterococcus faecium*, *Staphylococcus aureus*, *Klebsiella pneumoniae*, *Acinetobacter baumannii*, *Pseudomonas aeruginosa*, and Enterobacter species) group were characterized as of ‘priority status’ [14,15,16]. Through genetic mutations and the acquisition of mobile genetic elements (plasmids) [17], ESKAPE pathogens have developed resistance mechanisms against many antibiotic classes including oxazolidinones, lipopeptides, macrolides, fluoroquinolones, tetracyclines, and b-lactams and their combinations with b-lactamase inhibitors as well as last-line antibiotic groups such as carbapenems, glycopeptides, and polymyxins [18,19,20,21,22,23]. Additionally, resistance to lipoglycopeptides, although rare, has been reported recently [24].

Among European Union (EU) countries, Greece reports consistently higher-than-EU-average rates of resistance in the following hospital-acquired pathogens: *Acinetobacter baumannii*, *Klebsiella pneumoniae*, *Pseudomonas aeruginosa*, and *Enterococcus faecium* [6]. Carbapenem resistance in Enterobacterales causing hospital-acquired infections is 43.7% in our country, whereas other EU countries (Estonia, Finland, Iceland) report 0%. Particularly, a major problem seems to be the resistance of *Klebsiella pneumoniae* to third-generation cephalosporins and carbapenems (66.5% for Greece vs. 31.3% for EU and 58.3% vs. 7.9%, respectively), the resistance of *Pseudomonas aeruginosa* to carbapenems and fluoroquinolones (48.9% for Greece vs. 16.5% for EU and 46.8% vs. 18.9%, respectively), and the increasing multiresistance of *Acinetobacter baumannii* to cefepime, carbapenems, and aminoglycosides in both ICUs and clinical wards [5,6]. For example, the proportion of carbapenem-resistant *Acinetobacter baumannii* isolates in blood stream infections changed for meropenem from 79.7% and 95.3% in 2010 to 94.4% and 98.1% in 2017 in clinical wards and ICUs, respectively [6].

Our study estimated the antimicrobial resistance of clinical isolates in a 227-bed general hospital in Katerini, Greece. Katerini is the capital city of Pieria Regional Unit in Northern Greece. The municipal unit of Katerini has a population of 80,700 (according to the 2021 census) [25] and it is the second most populous urban area in the region after the city of Thessaloniki (the biggest city in Northern Greece). The General Hospital of Katerini is the only hospital in this regional unit and has approximately 2500 annual admissions.

Our findings concerning the susceptibility of bacterial strains of the ESKAPE pathogens to certain agents reveal some very interesting points. Regarding MRSA, a similar prevalence seemed to occur in our study and other published national data [6]. Ciprofloxacin resistance, although not stable, remains quite high [6]. It is encouraging that the susceptibility to glycopeptide (VAN, vancomycin; TEC, teicoplanin) and lipopeptide antibiotics (DAP, daptomycin) and oxazolidinones (LNZ, linezolid) is constantly high. As for Enterococci, *Enterococcus faecalis* was more sensitive than *Enterococcus faecium* to all antimicrobial classes except linezolid. However, the observed increasing level of *Enterococcus faecium* non-susceptibility to vancomycin and teicoplanin in our study, especially during 2021–2023, is problematic and highlights the need for close monitoring, although high resistance levels were reported from other EU countries as well [26].

In our study, *E. coli,* the most prevalent pathogen isolated from biological specimens, showed high susceptibility to many antibiotics, except amoxicillin/clavulanic acid and fluoroquinolones. Although resistance to the last-line group of antimicrobial agents (carbapenems, colistin) remained low during the whole 6-year period of testing, an increasing trend of non-susceptibility to third-generation cephalosporins and piperacillin/tazobactam is a serious problem since administration of an effective empirical antimicrobial treatment is essential for both hospitalized and community patients [6].

For *Enterobacter* species, because of the limited number of isolates during this 6-year surveillance study, more data on antimicrobial resistance need to be available in order to estimate certain conclusions. However, *Enterobacter cloacae* seems to have a much lower susceptibility to many antibiotic categories over time than *Enterobacter aerogenes*.

For *Klebsiella pneumoniae,* the major findings of this study are both the high resistance rates of isolates to all antibiotic classes, except amikacin (AMK) and colistin (COL), although a trend of increasing resistance to these agents is obvious, as well as a high proportion of multidrug resistance. Polemis et al. (2020) also reported high rates of carbapenem resistance [6].

Various Greek reports exist dealing with the phenomenon of high rates of carbapenem resistance among clinical isolates of *K. pneumoniae* since 2002. First, Verona integron-encoded Metallo-beta-lactamase (VIM) producers were the dominant ones during 2002–2007 [27] followed by the endemic presence of *K. pneumoniae* carbapenemase (KPC) producers since 2011 [28,29]. Afterwards, *K. pneumoniae*-producing New Delhi Metallo-beta-lactamase (NDM)-1 emerged in the country [30,31] and rapidly became the second most frequent carbapenemase encountered [32,33].

Regarding oxacillinase (OXA)-48-carbapenemases, since their entry in 2012, low rates of this carbapenemase class have been observed [33,34]. In our hospital, in the first study period (2018–2020), the dominant presence of KPC producers with a certain resistance phenotype (multiresistance with sensitivity only to amikacin and colistin) was the cause of the registered susceptibility to this agent. The increased amikacin resistance rate observed during the second study period (2021–2023) was due to the onset of NDM producers as well.

Regarding *Acinetobacter baumannii*, the extremely low susceptibility to all antibiotic categories, except colistin, and its predominance in hospital settings and especially in ICUs is an important finding of our study that has caused a great concern during the last years [5,6,35]. However, data on colistin-resistant/carbapenem-resistant *A. baumannii* isolates are more alarming and constitute a great challenge for both clinical practice and public health [36]. *A. baumannii* isolates in Greek hospitals produce almost exclusively the OXA-23 carbapenemase [37]. A limitation of this study is the determination of *A. baumannii* colistin susceptibility using automated system methods. Because strains resistant to colistin appear as false susceptible when tested automatically, both CLSI and EUCAST suggest the use of the broth microdilution method (BMD) as the gold-standard method [38].

A recent study that took place in our laboratory confirmed this finding, resulting in low accordance between these two methods (Vitek 2 and BMD), since 49.7% of A. baumannii strains were found sensitive to colistin using both Vitek 2 and BMD methods and 50.3% of strains were resistant to colistin when examined with the BMD method [39].

As for *Pseudomonas aeruginosa*, high rates of resistance to carbapenems and fluoroquinolones are in accordance with other national reports [5,6]. Additionally, similar resistance patterns to third- and fourth-generation cephalosporins (CAZ, ceftazidime; FEP, cefepime), piperacillin/tazobactam (PTZ), amikacin, and carbapenems exist between the two studied time periods. It seems that the decreased proportion of *Pseudomonas aeruginosa* VIM producers found during the last 20 years [40,41] combined with the increased presence of non-enzymatic mechanisms of carbapenem resistance [42,43] explain this phenomenon, as VIM producers exhibit multidrug-resistant phenotypes. Therefore, the low number of VIM producers could be responsible for the relatively low appearance of multidrug-resistant isolates observed in this study. Also very encouraging is the low resistance to colistin observed during these 6 years; this provides alternative therapeutic options to both hospital- and community-acquired infections caused by this microorganism.

Interesting findings about MDR resistance appeared during the pandemic years in our study. COVID-19 infection, with an estimated number of deaths higher than 6 million, has led to great morbidity and mortality worldwide [44]. Risk factors for developing COVID-19 in adults are considered as both demographic factors, such as older age, male sex, and ethnicity, and the presence of underlying diseases as well [44,45,46].

Additionally, reports demonstrate that COVID-19 patients are at greater risk of developing an MDR hospital-acquired infection due to their prolonged hospitalization, the increased rate of admissions to ICUs, and the need for mechanical ventilation [47,48]. Our data demonstrate that the significant increase in MDR pathogens in male patients (mainly in urine, blood, rectal, and pharyngeal specimens) observed in 2021–2023 when compared to 2018–2020 was caused mainly by admissions of patients with COVID-19 disease. In fact, older males (aged > 68 years) with mild and severe COVID-19 infection were the majority of hospital admissions during this time period, hospitalized in the ICU or either the internal medicine or COVID-19 clinical departments.

Another interesting finding of our study is the high rate of MDR pathogens in the community, as registered by the visits of outpatients to ambulatory and emergency care units of the hospital (11%). The explanation of this existing AMR reservoir, showing the spread of MDR pathogens from the hospital environment to the community, is probably the high antimicrobial consumption in our country. Greece has the highest percentage in consumption of antimicrobial agents among all European countries: in a 2020 ECDC report, antimicrobial consumption in defined daily doses (DDDs) per 100 inhabitants per day was 34.1 in Greece whereas the EU mean consumption was 19.4 DDDs per 100 inhabitants per day [5]. This overconsumption of antimicrobials may lead to the selection of resistant bacterial strains in Greece.

## 4. Materials and Methods

### 4.1. Study Design

This retrospective study was performed by analyzing the annual reports of the Laboratory of Microbiology in General Hospital of Katerini, which is a city in Northern Greece. The duration of the study was almost 6 years, starting from 1 January 2018 and ending on 30 June 2023. It consisted of two parts: in the first part of the study, we investigated the antimicrobial susceptibility of certain bacteria during the whole 6-year time period using the WHONET Greek electronic surveillance system for monitoring AMR in hospitals based on routine data. This time period was chosen because, from 1 January 2018 and onwards, data from the newly established ICU were included in our reports, making data from older periods incomparable. On the other hand, the CLSI breakpoints were used as cutoff values for antimicrobial resistance until 30 June 2023, making data from newer periods incomparable. In the second part of the study, we analyzed and compared the epidemiology and prevalence of multidrug-resistant (MDR) bacterial isolates between the time periods of 2018–2020 and 2021–2023.

The analysis was focused on the most important bacteria that were monitored globally and regionally within the surveillance networks: *Enterococcus faecalis* and *Enterococcus faecium*, *Staphylococcus aureus*, *Klebsiella pneumoniae*, *Acinetobacter baumannii*, *Pseudomonas aeruginosa*, *Escherichia coli*, *Enterobacter cloacae,* and *Enterobacter aerogenes*. The data were processed according to the institutional and national ethical standards and the World Medical Association (WMA) Declaration of Helsinki (1975), as revised in 2013. The protocol of the study was approved by the Ethics Committee of the General Hospital of Katerini (protocol code 8/30–8-2023).

The bacteria were isolated from biological products that were taken for diagnostic purposes from both hospitalized and community patients. Duplicate isolates were excluded from the study. Clinical isolates were identified by using classical bacteriology methods for microbial culture, on agar-based solid media, followed by biochemical testing of the obtained colonies performed by an automated Vitek 2 Compact system (Biomerieux, Lyon, France). Antibiotic sensibility was tested by using the minimal inhibiting concentration (MIC) method performed by the Vitek 2 Compact system. The interpretation of the antibiotic resistance was made according to CLSI (Clinical and Laboratory Standards Institute) guidelines (2022 edition).

MRSA (methicillin-resistant *Staphylococcus aureus*) strains were identified by using the Vitek 2 Compact Software based on an oxacillin non-susceptibility test for methicillin resistance. Multiple drug resistance (MDR) was defined as the antimicrobial resistance shown by a species of microorganism to at least one antimicrobial drug in three or more antimicrobial categories. Extensively drug resistance (XDR) was defined as the non-susceptibility of one bacteria species to all antimicrobial agents except in two or fewer antimicrobial categories. In conclusion, pan drug resistance (PDR) was defined as the non-susceptibility of bacteria to all antimicrobial agents in all antimicrobial categories [49].

The antibiotic classes that were used included penicillins (AMP, ampicillin; AMC, amoxicillin clavulanate), cephalosporins (FEP, cefepime; CTX, cefotaxime; CAZ, ceftazidime; TZP, piperacillin/tazobactam), carbapenems (IMP, imipenem; MEM, meropenem), fluoroquinolones (CIP, ciprofloxacin; LVX, levofloxacin), aminoglycosides (AMK, amikacin), sulfonamide (SXT, sulfamethoxazole/trimethoprim), and polymyxins (COL, colistin), and, furthermore, for the Gram-positive bacteria, glycopeptide antibiotics (VAN, vancomycin; TEC, teicoplanin), lipopeptide antibiotics (DAP, daptomycin), and oxazolidinones (LNZ, linezolid).

### 4.2. Statistical Analysis

Continuous variables are presented as a median and interquartile range, while categorical variables are described as counts and percentages. The comparison of the continuous values between the groups was performed with a nonparametric Mann–Whitney test while the comparison of the categorical variables was performed using Pearson chi-square or Fisher’s exact test. A univariate general linear (two-way ANOVA model) test was used to evaluate the association between age and sex in the two study periods. A Mann–Kendal test was performed for the trend analysis of the resistance to certain antibiotics regarding the ESKAPE pathogens during 2018–2023.

The SPSS 26.0 statistical package was used for the statistical analysis of the data, regarding the MDR rates, and *p* values < 0.05 were considered as statistically significant. The 5.6 software version of WHONET was used for the analysis of antimicrobial resistance rates of the isolates over time, making use of the CLSI 2021–2022 breakpoints. 

## 5. Conclusions

An overall increasing antimicrobial resistance rate of the dominant pathogens that are responsible for hospital-acquired infections (*Klebsiella pneumoniae*, *Acinetobacter baumannii*, *Pseudomonas aeruginosa,* and *Enterococcus faecalis*) over time has been registered. Multidrug resistance is more common in *Klebsiella pneumoniae* and *Acinetobacter baumannii* isolates, especially in the ICU. The frequency of VRE and MRSA isolated bacterial strains remains relatively low. The profile of the hospitalized patient with an MDR infection, mostly from urine-, blood-, and catheter-related specimens, is male, over 68 years, having COVID-19 at the peak of the pandemic years and admitted to the ICU or the department of internal medicine. Furthermore, the presence of pathogens with extensively drug resistance and pan drug resistance during the last 6 years is alarming. It is urgent to establish hospital-based antimicrobial stewardship and programs for infection prevention and control as well as educational programs on rational antimicrobial use and infection control for health care workers.

## Figures and Tables

**Figure 1 antibiotics-12-01595-f001:**
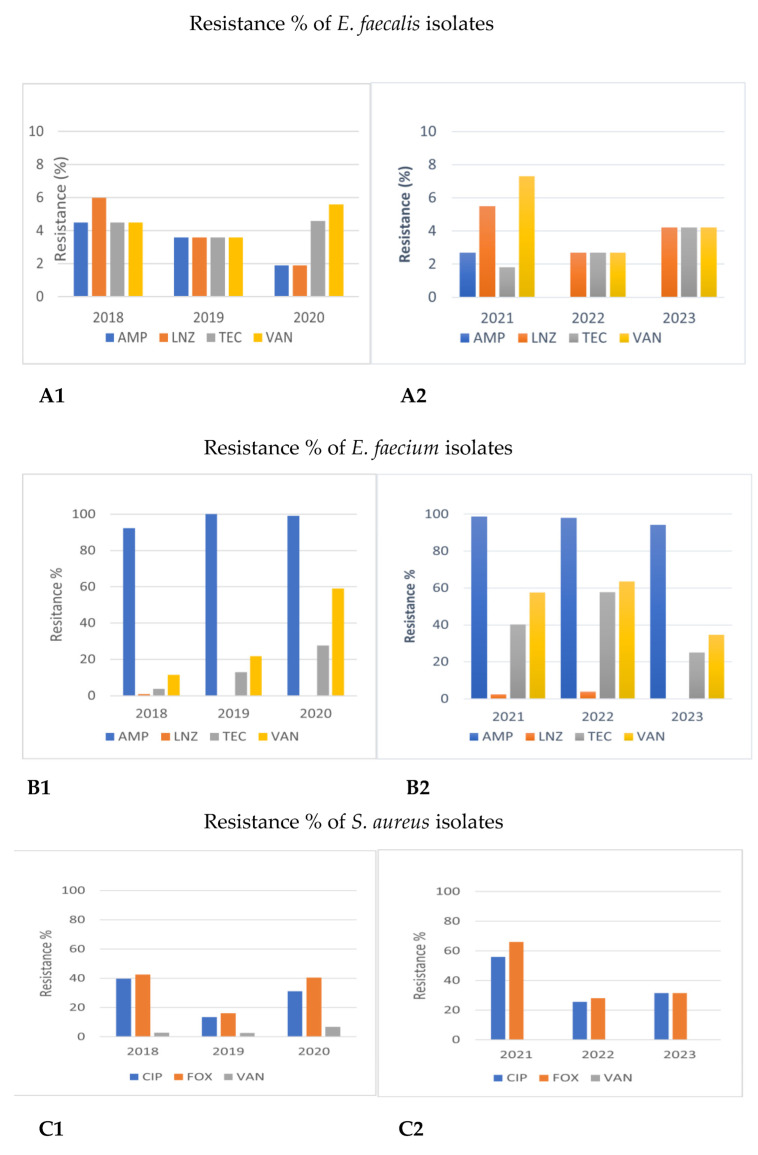
Comparison of antimicrobial resistance of Gram-positive cocci to certain antibiotics between 2018–2020 and 2021–2023. (**A1**,**A2**) Antimicrobial resistance of *Enterococcus faecalis* in 2018–2020 and 2021–2023. (**B1**,**B2**) Antimicrobial resistance of *Enterococcus faecium* in 2018–2020 and 2021–2023. (**C1**,**C2**) Antimicrobial resistance of *Staphylococcus aureus* in 2018–2020 and 2021–2023. AMP: ampicillin; LNZ: linezolid; TEC: teicoplanin; VAN: vancomycin; CIP: ciprofloxacin: FOΧ: cefoxitin.

**Figure 2 antibiotics-12-01595-f002:**
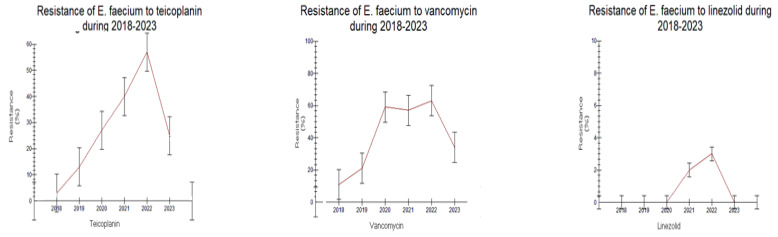
Trend of *Enterococcus faecium* resistance to antibiotics during the study period 2018–2023.

**Figure 3 antibiotics-12-01595-f003:**
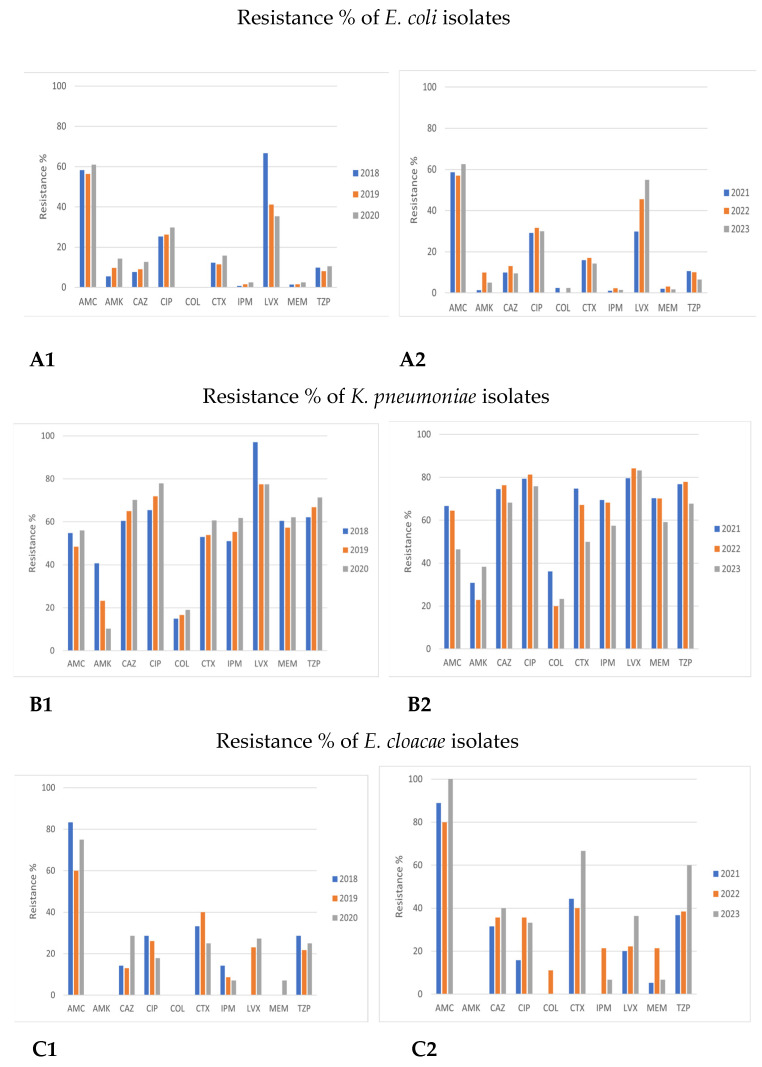
Comparison of antimicrobial resistance of Enterobacterales species to certain antibiotics between 2018–2020 and 2021–2023 time periods. (**A1**,**A2**) Antimicrobial resistance of *E. coli* in 2018–2020 and 2021–2023. (**B1**,**B2**) Antimicrobial resistance of *Klebsiella pneumoniae* in 2018–2020 and 2021–2023. (**C1**,**C2**) Antimicrobial resistance of *Enterobacter cloacae* in 2018–2020 and 2021–2023. (**D1**,**D2**) Antimicrobial resistance of *Enterobacter aerogenes* in 2018–2020 and 2021–2023.

**Figure 4 antibiotics-12-01595-f004:**
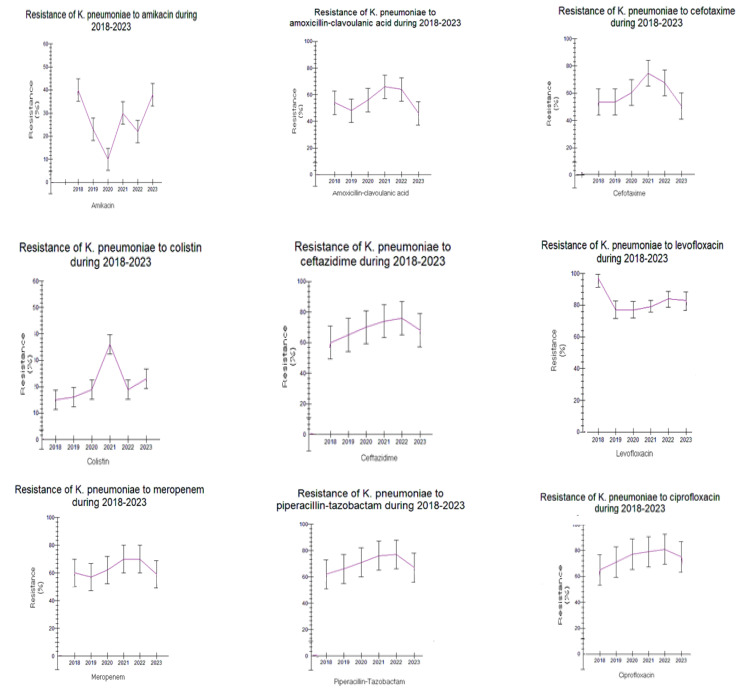
Trend of *Klebsiela pneumoniae* resistance to antibiotics during the study period 2018–2023.

**Figure 5 antibiotics-12-01595-f005:**
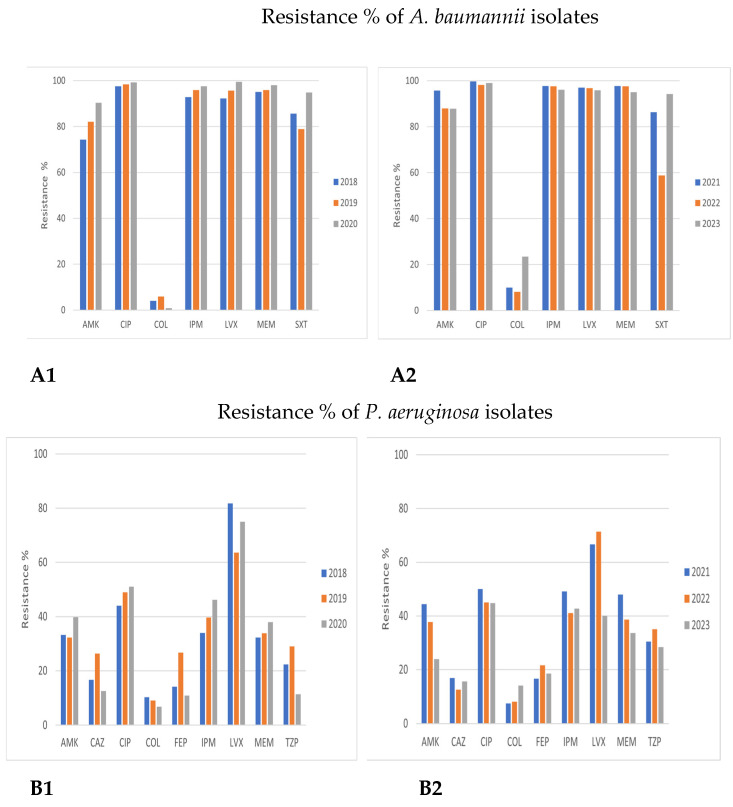
Comparison of antimicrobial resistance of *Acinetobacter baumannii* and *Pseudomonas aeruginosa* to certain antibiotics in 2018–2020 and 2021–2023. (**A1**,**A2**) Antimicrobial resistance of *Acinetobacter baumannii* in 2018–2020 and 2021–2023. (**B1**,**B2**) Antimicrobial resistance of *Pseudomonas aeruginosa* in 2018–2020 and 2021–2023.

**Figure 6 antibiotics-12-01595-f006:**
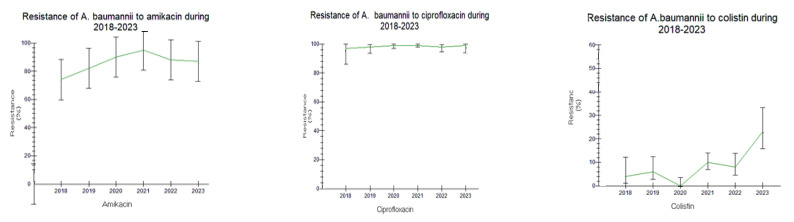
Trend of *Acinetobacter baumannii* resistance to antibiotics during the study period 2018–2023.

**Figure 7 antibiotics-12-01595-f007:**
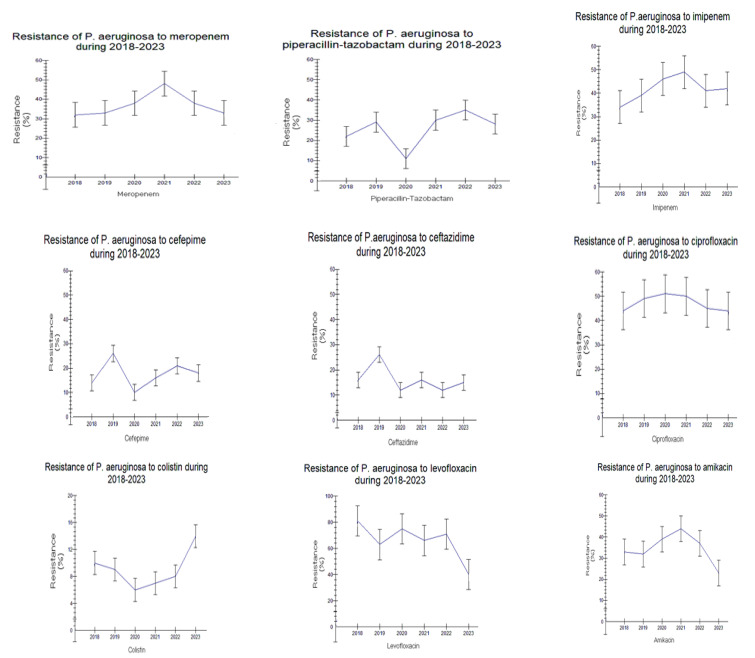
Trend of *Pseudomonas aeruginosa* resistance to antibiotics during the study period 2018–2023.

**Figure 8 antibiotics-12-01595-f008:**
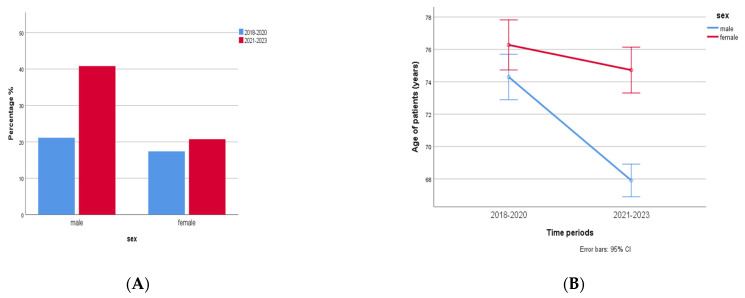
(**A**) MDR in males and females in 2018–2020 (n = 1094) and 2021–2023 (n = 673). A significant increase (2 fold) in MDR bacteria was observed in males compared to females in the time period 2021–2023 compared to the time period 2018–2020 (*p* < 0.0001, r = −0.115). (**B**) Cases with MDR isolates in men were observed in younger age during the second time period, 2021–2023, compared to 2018–2020 (two-way ANOVA, *p* < 0.0001, r = 0.062).

**Figure 9 antibiotics-12-01595-f009:**
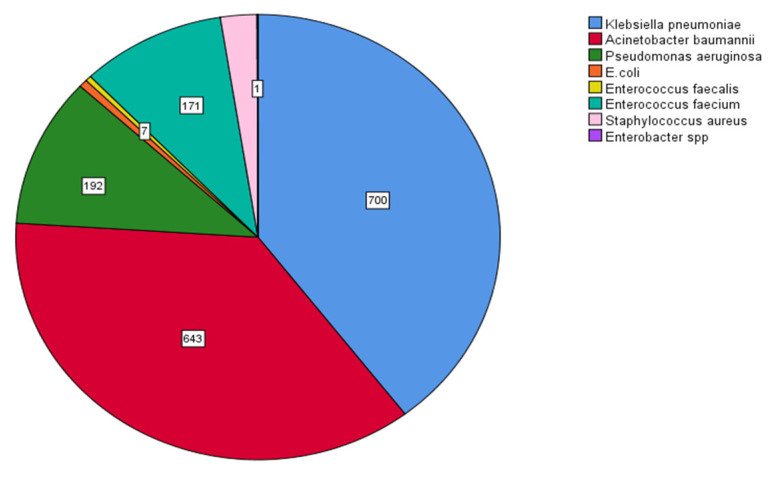
Prevalence of multidrug-resistant (MDR) bacteria. MDR *Klebsiella pneumoniae* (700/1767, 39.6%), *Acinetobacter baumannii* (643/1767, 36.4%), *Pseudomonas aeruginosa* (192/1767, 10.9%), and *Enterococcus faecium* (171/1767, 9.7%).

**Figure 10 antibiotics-12-01595-f010:**
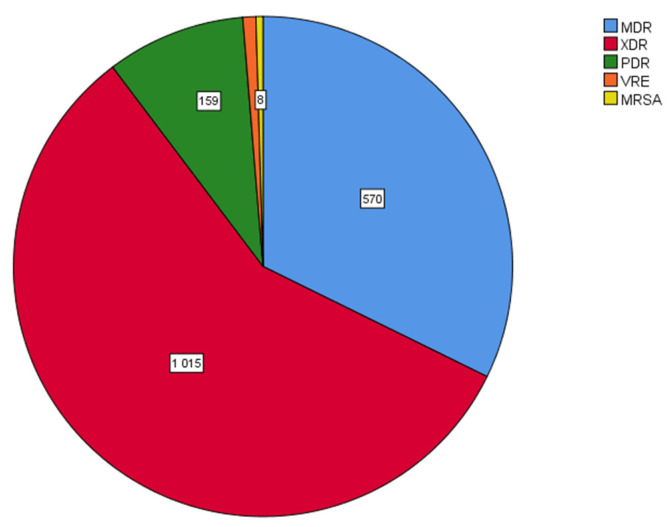
Distribution of bacterial clinical isolates according to the kind of antimicrobial resistance. XDR isolates were the most prevalent ones (1015/1767, 57.4%), followed by MDR strains (570/1767, 32.3%) and PDR strains (159/1767, 9%). The prevalence of VRE and MRSA isolated bacterial strains was low. MDR: multidrug resistant; XDR: extensively drug resistant; PDR: pan drug resistant; VRE: vancomycin-resistant enterococci; MRSA: methicillin-resistant *Staphylococcus aureus*.

**Figure 11 antibiotics-12-01595-f011:**
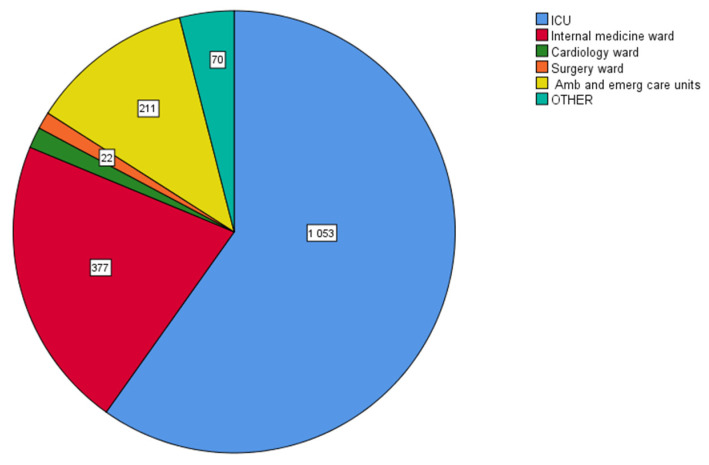
Distribution of multidrug-resistant (MDR) clinical isolates in various hospital departments. Intensive care unit (ICU) harbored the majority of MDR isolates (1053/1760, 59.8%), followed by the department of internal medicine (377/1760, 21.4%) and the hospital ambulatory and emergency care units (211/1760, 11%).

**Figure 12 antibiotics-12-01595-f012:**
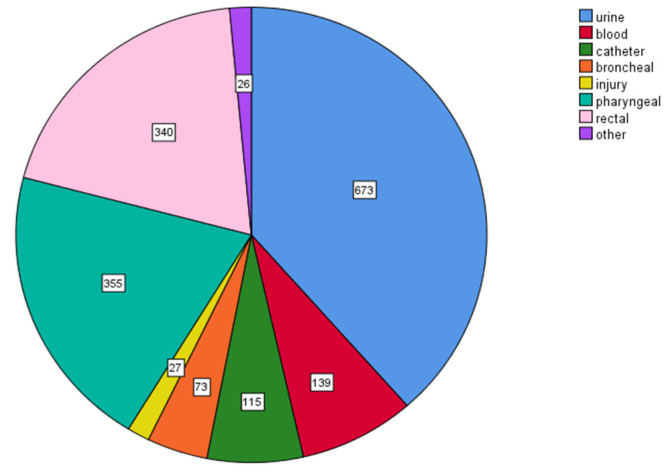
Distribution of multidrug-resistant (MDR) clinical isolates according to specimen type. Most MDR isolates were obtained from urine cultures (673/1748, 38.5%), from pharyngeal (355/1748, 20.3%) and rectal specimens (340/1748, 19.5%), and from blood- and catheter-related cultures (254/1748, 14.6%).

**Figure 13 antibiotics-12-01595-f013:**
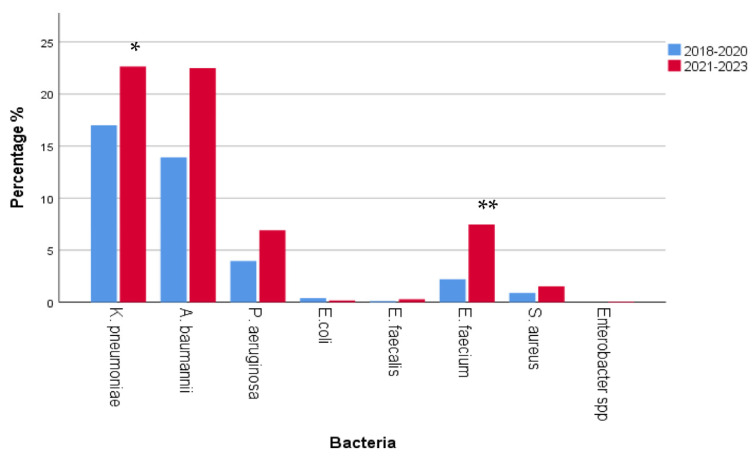
Comparison of multidrug resistance (MDR) prevalence between 2018–2020 and 2021–2023. A significant rise in *Klebsiella pneumoniae* and *Enterococcus faecium* MDR isolates was observed (*K. pneumoniae* * *p* = 0.002, r = 0.073; *E. faecium* ** *p* < 0.0001, r = 0.105). However, a non-significant rise in *Acinetobacter baumannii* and *Pseudomonas aeruginosa* MDR isolates was also observed (*p* > 0.05).

**Figure 14 antibiotics-12-01595-f014:**
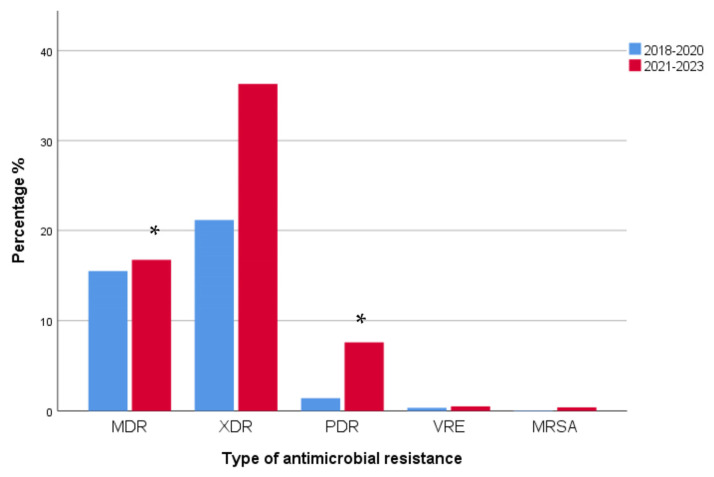
Distribution of isolated bacterial strains according to the type of antimicrobial resistance in 2018–2020 and 2021–2023. MDR and PDR strains increased significantly during 2021–2023 when compared to 2018–2020 (MDR, * *p* < 0.0001, r = 0.136; PDR * *p* < 0.0001, r = 0.147). A non-significant rise in XDR strains in 2021–2023 when compared to these in 2018–2020 was also observed (*p* > 0.05). MDR: multidrug resistant; XDR: extensively drug resistant; PDR: pan drug resistant; VRE: vancomycin-resistant enterococci; MRSA: methicillin-resistant *Staphylococcus aureus*.

**Figure 15 antibiotics-12-01595-f015:**
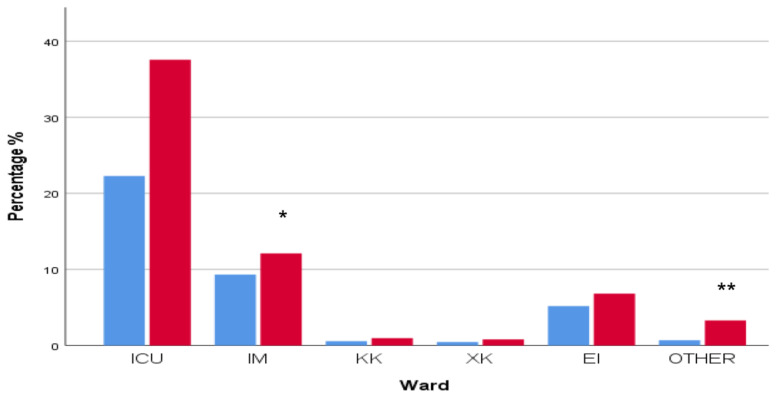
Distribution of multidrug-resistant (MDR) isolated bacterial strains in various hospital departments in 2018–2020 and 2021–2023. Significant increases in MDR cases were observed mainly in internal medicine (* *p* < 0.05, r = 0.054) and other departments (** *p* = 0.0001, r = 0.089). A larger but non-significant increase in MDR cases was also observed in ICU (*p* > 0.05). ICU: intensive care unit; IM: internal medicine; KK: cardiology department; XK: surgery department; EI: ambulatory and emergency care units; OTHER: other departments.

**Table 1 antibiotics-12-01595-t001:** Prevalence of the examined isolates per year.

Bacterial Strains	2018 No.	2019 No.	2020 No.	2021 No.	2022 No.	2023 No.	Total No.	Mean % *
*Enterococcus faecalis*	69	60	111	113	77	25	455	3.7
*Enterococcus faecium*	33	48	106	176	106	54	523	4.3
*Staphylococcus* *aureus*	70	38	76	60	39	19	302	2.5
*Escherichia* *coli*	1033	954	779	759	651	409	4585	37.4
*Klebsiella* *pneumoniae*	366	377	391	563	302	236	2235	18.2
*Enterobacter cloacae*	15	15	8	25	33	20	116	0.95
*Enterobacter aerogenes*	9	14	9	6	2	14	54	0.44
*Acinetobacter baumannii*	86	127	253	314	172	106	1058	8.6
*Pseudomonas aeruginosa*	228	145	208	274	174	97	1126	9.2
Total	2205	2140	2321	2658	1800	1150	12,274	100

* %: prevalence per year.

**Table 2 antibiotics-12-01595-t002:** Distribution of the most frequent bacterial strains according to biological products analyzed during 2018–2023.

Cultures	2018 No.	2019 No.	2020 No.	2021 No.	2022 No.	2023 No.
Urine	1714 (78%)	1582 (75%)	1498 (65%)	1658 (63%)	1293 (72%)	807 (71%)
Blood	124 (6%)	144 (7%)	229 (10%)	260 (10%)	156 (9%)	111 (10%)
Pharyngeal	53 (2%)	78 (4%)	128 (6%)	175 (7%)	89 (5%)	52 (5%)
Rectal	53 (2%)	70 (3%)	138 (6%)	193 (7%)	68 (4%)	51 (4%)
Catheter	27 (1%)	46 (2%)	100 (4%)	121 (5%)	38 (2%)	28 (2%)
Bronchial	46 (2%)	64 (3%)	56 (2%)	63 (2%)	13 (1%)	28 (2%)

## Data Availability

Data supporting the reported results are kept by the authors and are available on demand. Patient consent was waived, because this was a retrospective study concerning data reported to WHONET; the protocol of the study was approved by the Ethics Committee of the General Hospital of Katerini (protocol code 8/30–8-2023) on the basis of publishing no private data of the patients.

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
