# Peer review of "Surveillance of Antimicrobial Resistance and Multidrug Resistance Prevalence of Clinical Isolates in a Regional Hospital in Northern Greece"

_antibiotics, 2023, doi:10.3390/antibiotics12111595_

Round 1
Reviewer 1 Report
Comments and Suggestions for Authors
The authors used the data of clinical isolates in a regional hospital in Northern Greece to evaluate the prevalences changes of DR during the past 6 years.
Major concern:
The authors have to reconsider the analytic plan. The decision to arbitrarily categorize the six years into 2018-2020 and 2021-2023 is not appropriate, given a substantial variation in drug resistance prevalence from year to year. It may be more appropriate to use a trend test or a similar approach to assess the trend in antimicrobial resistance (AMR) over the past 6 years. The same issue applies to the results presented in Figure 4. The research question for Figure 4 was whether the trend in MDR prevalence over the past six years varied (effectively modified) by gender. A more suitable analytic plan should align with this study question. Furthermore, a lot of comparisons that can be done in the same multivariate models were performed independently (e.g. the results of figures 5-9). Lastly, all "significant" findings should be accompanied by corresponding effect sizes and p-values or 95% confidence intervals (CIs). I recommend consulting a statistical expert to address these concerns and ensure proper statistical analysis.
Minor concerns
1. The presentation of the results requires a significant revision. Figures 1-4 assess whether resistance prevalences vary by year, while figures 5-9 examine resistance prevalence across several factors. However, starting from figure 10, the authors revert to questions about variations by year, creating an awkward transition in the flow.
2. Line 17. “ (63-78 %)”. While I later figured out that the range here refers to the data range during the past six years, but it was not clear when I first read it.
3. Line 58. This study provided information about the prevalence of resistance during the past 6 years. However, this study did not provide any evidence on “effective antimicrobial treatment”.
4. Line 68 pandemic years may refer to Covid-19. But it needs further clarification.
5. Line 75. A majority of this study was about the comparison between 2018-2020 and 2021-2023. It reads random when the topic was suddenly switched to things happened during pandemic.
6. Lines 80-81. First, whether the urine samples significantly decreased during the two years need a test. Second, I expected to see a description about other types of samples increased due to Covid-19 to echo the following description “The number of urine strains was decreased during the years 2020 - 2021, reflecting the dynamics of the pandemic years of 2020 and 2021, when the majority of patients had COVID-19 infection.”
7. Table 1. The footnote “frequency per year” for “Mean “was not clear.
8. It is not intuitive that in the bar plots, the three bars labelled with the same drug names referred to the results of 2018, 2019, and 2020. Please label them clearly.
9. Lines 119-120. Most of the results in this paragraph focused on whether things varied by time, but the contents in lines 119-120 suddenly switched to a comparison between two bacterial. It reads very strange.
10. Line 121 VRE in the content, but VAN was used in the figures
11. Line 329 AMR need to be spelled out.
Comments on the Quality of English LanguageMinor editing of English language required
Author Response
Answer to Reviewer 1.
Major concern:
The authors have to reconsider the analytic plan. The decision to arbitrarily categorize the six years into 2018-2020 and 2021-2023 is not appropriate, given a substantial variation in drug resistance prevalence from year to year. It may be more appropriate to use a trend test or a similar approach to assess the trend in antimicrobial resistance (AMR) over the past 6 years. The same issue applies to the results presented in Figure 4. The research question for Figure 4 was whether the trend in MDR prevalence over the past six years varied (effectively modified) by gender. A more suitable analytic plan should align with this study question. Furthermore, a lot of comparisons that can be done in the same multivariate models were performed independently (e.g. the results of figures 5-9). Lastly, all "significant" findings should be accompanied by corresponding effect sizes and p-values or 95% confidence intervals (CIs). I recommend consulting a statistical expert to address these concerns and ensure proper statistical analysis.
Answer to Major concern:
Thank you very much for this comment that gave us the opportunity to make major revisions in our manuscript according to your suggestions.
As stated in lines 462-466 of the initial manuscript (corresponding to lines 482-490 of the revised manuscript), “In the first part of the study we investigated the antimicrobial susceptibility of certain bacteria during the whole six year time period while in the second part of the study we analyzed and compared the epidemiology and prevalence of multidrug resistant (MDR) bacterial isolates between 2018-2020 and 2021-2023”. These two time-periods were not selected arbitrarily, as – after studying resistance during the 6 year period – we noticed differences in MDR bacteria between the two time periods that were investigated in the second part of the study.
In the first part of the study we investigated the whole 6 year time period year by year; these results are presented in detail in Figures 1, 2 and 3 of the initial manuscript. After your suggestion, we investigated specially the trend in antimicrobial resistance (AMR) over the past 6 years and we have added 4 new figures (Figures 2, 4, 6 and 7) in the revised manuscript (the former Figures 1, 2, 3 have been adapted and correspond now to Figures 1, 3 and 5 in the revised text).
Concerning your comment on Figure 4, we investigated the trend in MDR prevalence over the past six years according to gender. The results are presented in an additional figure (part B of the Figure 8 that corresponds now to Figure 4).
We have consulted a statistical expert and, according to his opinion, the comparison of the categorical variables was performed using Pearson chi square or Fisher's exact test. Finally, we have added p values and effect sizes in all findings.
All changes and additional text in Figures and in the manuscript are marked in red color.
Minor concerns
- The presentation of the results requires a significant revision. Figures 1-4 assess whether resistance prevalences vary by year, while figures 5-9 examine resistance prevalence across several factors. However, starting from figure 10, the authors revert to questions about variations by year, creating an awkward transition in the flow.
As we have stated previously (lines 462-466 of the initial manuscript), “In the first part of the study we investigated the antimicrobial susceptibility of certain bacteria during the whole six year time period while in the second part of the study we analyzed and compared the epidemiology and prevalence of multidrug resistant (MDR) bacterial isolates between 2018-2020 and 2021-2023”. We do not revert to questions about variations by year (this is studied in the 1st part, the variation of antimicrobial resistance to each antibiotic by year; in the 2nd part of the study we investigated something different: comparison of MDR, XDR and PDR between the two time periods 2018-2020 and 2021-2023).
- Line 17. “ (63-78 %)”. While I later figured out that the range here refers to the data range during the past six years, but it was not clear when I first read it.
According to your comment, we incorporated the appropriate clarification in line 17 in red color.
- Line 58. This study provided information about the prevalence of resistance during the past 6 years. However, this study did not provide any evidence on “effective antimicrobial treatment”.
The standard guidelines and the appropriate antimicrobial protocols are followed for the optimal antimicrobial treatment in our hospital. Nevertheless, as antimicrobial resistance is higher in Greece than in other European countries, in order to achieve an effective antimicrobial treatment, the standard antimicrobial guidelines in Greece are different from the international guidelines. So, the issue of the rational and effective antimicrobial treatment in Greece is rather complicated and it’s out of the scope of this study. Nevertheless, knowledge on antimicrobial resistance of clinical strains may lead to a more effective and successful antimicrobial treatment.
- Line 68 pandemic years may refer to Covid-19. But it needs further clarification.
Thank you for this comment. We added the appropriate clarification in the text.
- Line 75. A majority of this study was about the comparison between 2018-2020 and 2021-2023. It reads random when the topic was suddenly switched to things happened during pandemic.
In fact, most of this study was about the comparison between 2018-2020 and 2021-2023. Nevertheless, due to COVID 19 pandemic, an increase in the number of hospitalized patients lead to a corresponding increase in the number of main pathogens causing hospital acquired infections (Klebsiella pneumoniae, Acinetobacter baumannii, Pseudomonas aeruginosa) in 2021, as it is depicted in Table 1. This change was worth mentioning.
- Lines 80-81. First, whether the urine samples significantly decreased during the two years need a test. Second, I expected to see a description about other types of samples increased due to Covid-19 to echo the following description “The number of urine strains was decreased during the years 2020 - 2021, reflecting the dynamics of the pandemic years of 2020 and 2021, when the majority of patients had COVID-19 infection.”
In our hospital, a great part of urine specimens come from community outpatients. Although total urine specimens in 2021 were much lower (63% of total specimens) than in 2018 (78% of total specimens), the true decrease in urine specimens from outpatients is much higher than the number presented in Table 2, due to the simultaneous increase of hospitalized patients due to COVID 19 in 2021.
According to your suggestion, we rephrased lines 80-81 as follows: “The number of urine strains (deriving mainly from community specimens) was decreased during the peak of COVID 19 pandemic in 2021, while blood, pharyngeal, rectal and catheter specimens were increased, reflecting the long hospitalization due to COVID-19 infection.
- Table 1. The footnote “frequency per year” for “Mean “was not clear.
Thank you for this comment. By mistake it was written “frequency per year”. We have replaced the world “frequency” with the word “prevalence” per year. We have also made the proper changes in the graphs.
- It is not intuitive that in the bar plots, the three bars labelled with the same drug names referred to the results of 2018, 2019, and 2020. Please label them clearly.
According to your suggestion, we have replaced the 3 figures (Figures 1, 2, 3 in the initial text) with Figures 1, 3, 5 in the revised text, in which we have labeled the years for each antibiotic.
- Lines 119-120. Most of the results in this paragraph focused on whether things varied by time, but the contents in lines 119-120 suddenly switched to a comparison between two bacterial. It reads very strange.
You are right, most of the results in this paragraph focused on whether things varied by time. Nevertheless, resistance of E. faecalis and E. faecium to linezolid in both time periods caught our attention, because we observed an interesting phenomenon, that is opposite to what is usually observed: Usually, resistance to linezolid is higher in E. faecium than in E. faecalis; in our work we observed the opposite: resistance to linezolid was higher in E. faecalis than in E. faecium. As this phenomenon is unexpected, we thought that it is worth mentioning.
- Line 121 VRE in the content, but VAN was used in the figures
VRE (Vankomycin Resistant Enterococcus) in the content refers to Enterococcus that is resistant to vancomycin. In the figures we represent resistance of Enterococcus to antibiotics: VRE is resistant to VAN (vancomycin)
- Line 329 AMR need to be spelled out
Spell out was done (line 349 in the revised text).
The manuscript has been under proofreading by a native English speaker.
All important changes in the manuscript – except some minor changes in grammar, syntax and spelling – appear in the text in red color.
Reviewer 2 Report
Comments and Suggestions for Authors
The study entitled “Surveillance of antimicrobial resistance and multidrug resistance prevalence of clinical isolates in a regional hospital in Northern Greece” is well described and discussed. However, several questions need to be addressed before final acceptance.
1. To compare antibiotic resistance of the isolates from two different periods, I recommend merging the graph (figure 3) plotting the grouped data, and performing the statistical significance between the resistance in two different time frames against each antibiotic.
2. I suggest describing the method used to estimate the frequency percentage against the chosen antibiotics.
3. More importantly the percentage of frequency was calculated against which antibiotics? Also, how many times it has been evaluated? Why there is no error in the graph?
4. Yes, I agree that most of the Acinetobacter Spp. seems to have resistance against multiple antibiotics. A similar study has also supported your study, which is worth citation (DOI: 10.1016/j.micpath.2020.104287).
5. I strongly suggest to improve the methodology section of your manuscript.
6. Please describe the method used to identify the bacterial species.
7. The legends of Figure 4 and 9 indicates that statistical analysis has been done on these data, however, it is not indicated in the figures. Please cross-check the analyses and indicate the significance level in the plotted graph.
Author Response
Answers to Reviewer 2
- To compare antibiotic resistance of the isolates from two different periods, I recommend merging the graph (figure 3) plotting the grouped data, and performing the statistical significance between the resistance in two different time frames against each antibiotic.
Thank you for this comment. According to your suggestion, we haveplotted the grouped data of figure 3 infigures6 and 7, depicting the trend of antimicrobial resistance of Acinetobacter baumanii and Pseudomonas aeruginosa in the revised document. Additionally, we have added two more figures of grouped data (figures 2 and 4), depicting the trend of antimicrobial resistance of E. faecium and K. pneumoniae. Accordingly, we have renumbered the figures 1, 2 and 3 of the initial document as figures 1, 2 and 5 in the revised document.
Although there is a trend of increase in antibiotic resistance year by year with a peak in 2022, there is no statistical significance in resistance in the two different time frames. We must note here that as stated in lines 462-466 of the initial manuscript (corresponding to lines 482-490, “In the first part of the study we investigated the antimicrobial susceptibility of certain bacteria during the whole six year time period while in the second part of the study we analyzed and compared the epidemiology and prevalence of multidrug resistant (MDR) bacterial isolates between 2018-2020 and 2021-2023”. These two time-periods were not selected arbitrarily, as – after studying resistance during the 6 year period – we noticed differences in MDR, XDR and PDR bacteria between the two time periods that were investigated in the second part of the study. Data on statistical significance, p values and effect size are presented in detail in the part 2.2 of the study and are depicted in Figures 9-11 in the initial document and Figures 13-15 in the revised document.
- I suggest describing the method used to estimate the frequency percentage against the chosen antibiotics.
Thank you for this comment. By mistake it was written “frequency per year”. We have replaced the world “frequency” with the word “prevalence” per year.
- More importantly the percentage of frequency was calculated against which antibiotics? Also, how many times it has been evaluated? Why there is no error in the graph?
Thank you for this comment. By mistake it was written “frequency per year”. We have replaced the world “frequency” with the word “prevalence” per year. We have also made the proper changes in the graphs.
The antibiotics checked are mentioned in lines 492-495 and 518-525 of the revised document (467-470 and 492-499 of the initial document).
Standard Error is depicted in Figure 8 of the revised document (Figure 4 in the initial document). In other figures variables are nominal (resistant or non-resistant), and we have number and percentage of resistant strains.
- Yes, I agree that most of the Acinetobacter Spp. seems to have resistance against multiple antibiotics. A similar study has also supported your study, which is worth citation (DOI: 10.1016/j.micpath.2020.104287).
Thank you for this comment. The study that you recommended is a very interesting one. Although it deals with Acinetobacter courvalinii, we decided to include it in our references as an evidence of the great concern caused by the increase of antimicrobial resistance of Acinetobacter species during the last years.
- I strongly suggest to improve the methodology section of your manuscript.
Thank you for this comment. We have improved the section of methodology in our manuscript, according to your suggestion (see lines 483 to 489 and 532 to 538 in the revised text; changes are marked in red color)
- Please describe the method used to identify the bacterial species.
The method is described in lines 500-511 of the revised document (475-485 in the initial document).
In more detail, routine laboratory techniques performed in order to identify the bacterial species were:
- Inspection (look at the microorganism through a microscope)
- Identification (phenotypic to our laboratory)
- Inoculation (The sample is placed into a container of sterile medium (we use solid agar plates) that provides microbes with the appropriate nutrients to sustain growth)
- Incubation (An incubar can be used to adjust the proper growth conditions of a sample)
- Isolation (The end result of inoculation and incubation is isolation of the microbe.)
- After the isolation of the microbes we use morphological (gram stain, motility, appearance of cultures) and biochemical techniques to identify the microorganisms. In our laboratory the biochemical techniques are performed by Vitek 2 automated System (Biomerieux).
- Antibiotic sensibility is tested by using the minimal inhibiting concentration
(MIC) method performed by Vitek 2 Compact system, as well. The
interpretation of the antibiotic resistance was made according to CLSI (Clinical
and Laboratory Standards Institute) guidelines (2022th edition) until 30th June
- Ever since, CLSI guidelines have been replaced by EUCAST guidelines.
However, these details in methods cannot be reported in detail in the text, because they are well-known and established routine practices.
- The legends of Figure 4 and 9 indicates that statistical analysis has been done on these data, however, it is not indicated in the figures. Please cross-check the analyses and indicate the significance level in the plotted graph.
We have done it in Figure 4 (Figure 8 in the revised document). Due to technical problems we could not do it in Figure 9 (Figure 13 in the revised document); nevertheless, statistical data are included in the legend and in the text.
The manuscript has been under proofreading by a native English speaker.
All important changes in the manuscript – except some minor changes in grammar, syntax and spelling – appear in the text in red color.
Round 2
Reviewer 1 Report
Comments and Suggestions for Authors
1. Some of my major concerns were either neglected or not taken care of seriously.
a. "Furthermore, many comparisons that could be done in the same multivariate models were performed independently (e.g., the results of figures 5-9)." This comment was neglected.
b. In scientific writing, unless the authors aimed to provide descriptive analyses, all comparisons with terms like "higher," "rise," or "oblivious trend" need a formal test. This applies to lines 121, 125, 133, and 165-168, as well as 173.
c. The authors proposed to compare resistant patterns from 2018-2020 and 2021-2023. In the figures, I observed that the prevalences of resistance varied greatly within 2018-2020 and within 2021-2023. Therefore, it may not be appropriate to lump the three-year data together. I acknowledge that the authors provided figures to show the trend of change during the six years. However, what I meant was to perform a trend test, considering the data from each year as an independent unit.
d. Figure 8B has now replaced Figure 4. However, I am not sure how it connects with the description shown in lines 241-256, especially the y-axis labeled as "age," which is confusing.
2. The resolution of many figures is very poor. I cannot even read some of the text shown in the figures. For the figures showing trend across six year, the CIs overlapped with the main title.
3. For Figure 13 and 14, the y-axis should display proportions rather than absolute numbers. I am also unsure why only one p-value was provided given a significant rise in many drugs.
4. Figure 15: It is not appropriate to compare the raw case numbers, as the total case number for 2023 cannot be complete (it is only October 2023). I am also unsure why only one p-value was provided given a significant rise in many drugs. I noticed 'PK' in the figure, but it appears to represent 'IM'.
Author Response
Answer to 2nd round Review
- Some of my major concerns were either neglected or not taken care of seriously. a."Furthermore, many comparisons that could be done in the same multivariate models were performed independently (e.g., the results of figures 5-9)." This comment was neglected.
We have not neglected this comment. We have answered it in the 5th paragraph of our “Answer to Major concern”:
We have consulted a statistical expert and, according to his opinion, the comparison of the categorical variables was performed using Pearson chi square or Fisher's exact test.
The expert opinion of our statistician is that a multivariate model would not be proper for the comparison of categorical variables such as those in figures 5-9 (9-13 in the revised document). Instead, he proposed the Pearson chi square or Fisher's exact test as the optimal statistical models.
b. In scientific writing, unless the authors aimed to provide descriptive analyses, all comparisons with terms like "higher," "rise," or "oblivious trend" need a formal test. This applies to lines 121, 125, 133, and 165-168, as well as 173.
Thank you for this comment.
After conducting a trend test, we found no statistical difference in any trend of resistance to the examined drugs of all Gram positive cocci during the whole study period 2018-2023, except for resistance of E. faecalis to ampicillin. Please check the text 9lines 121-138 in the newly revised document; changes are marked in red color).
Moreover, no significant changes in trends were observed, considering the drug resistance of Gram negative bacteria, Enterobacterales, Acinetobacter baumannii and Pseudomonas aeruginosa, as well, during 2018-2023, except for a significant rise of Klebsiella pneumoniae resistance to colistin during the six year period. We made some changes in the text, according to your suggestions (please check lines 174-180, 220-224 and 231-234 in the newly revised document.
c. The authors proposed to compare resistant patterns from 2018-2020 and 2021-2023. In the figures, I observed that the prevalences of resistance varied greatly within 2018-2020 and within 2021-2023. Therefore, it may not be appropriate to lump the three-year data together. I acknowledge that the authors provided figures to show the trend of change during the six years. However, what I meant was to perform a trend test, considering the data from each year as an independent unit.
Thank you for your comment. According to your suggestion, we conducted a Mann-Kendal trend test for the analysis of resistance to various antibiotics of the ESKAPE pathogens for each year of the studied period 2018-2023.
d. Figure 8B has now replaced Figure 4. However, I am not sure how it connects with the description shown in lines 241-256, especially the y-axis labeled as "age," which is confusing.
Figure 4 has been substituted by Figure 4a, and a new figure (Figure 8b) was added according to your previous suggestions. Concerning figure 8b, we substituted the label “age” in y-axis with the label “Age of patients” in the newly revised text. In figure 8b the blue line connects the mean age of men with MDR isolates in the first study period (2018-2020) with the mean age of men with MDR isolates in the second study period (2021-2023). As we have explained in the text (lines 262-265) and in the Figure 4 legend (lines 282-284 of the revised manuscript), Cases with MDR isolates in men were observed in younger age during the second time period 2021-2023 compared to 2018-2020 and this difference was statistically significant (two-way ANOVA, p<0.0001, r= 0.062).
- The resolution of many figures is very poor. I cannot even read some of the text shown in the figures. For the figures showing trend across six year, the CIs overlapped with the main title.
Thank you very much for your comment. According to your suggestion, we have improved the resolution of Figures 1-7, making effort for no overlapping between Cls and the main titles in trend graphs (see Figure 6).
- For Figure 13 and 14, the y-axis should display proportions rather than absolute numbers. I am also unsure why only one p-value was provided given a significant rise in many drugs.
Thank you for your comment. We have changed absolute numbers to proportions in y axis of both Figures 13 and 14, according to your suggestions. We also provided all p-values and changed the text in lines 329-335, 356-358 and in the legends of Figures 13 and 14 (lines 350-353 and 366-369 in the newly revised document).
- Figure 15: It is not appropriate to compare the raw case numbers, as the total case number for 2023 cannot be complete (it is only October 2023). I am also unsure why only one p-value was provided given a significant rise in many drugs. I noticed 'PK' in the figure, but it appears to represent 'IM'.
The duration of our study was almost six years, starting from January 1st 2018 and ending on June 30th 2023. This time period was chosen because from January 1st 2018 and onwards, data from the newly established ICU were included in our reports to WHONET, making data from older periods incomparable. On the other hand, the CLSI break-points were used as cut off values for antimicrobial resistance until June 30th 2023. Ever since, the CLSI guidelines have been replaced by EUCAST guidelines, making data from newer periods incomparable.
We have provided all p-values in the text (lines 374-381) and in the legend of Figure 15 (lines 385-388 in the revised document). ‘PK’ in the figure was substituted with ‘IM’.
All changes in the manuscript are marked in red color.
